# Effects of Nitrification Inhibitors on Soil Nitrification and Ammonia Volatilization in Three Soils with Different pH

**Lei Cui** [1,2], **Dongpo Li** [1,*], **Zhijie Wu** [1], **Yan Xue** [1], **Furong Xiao** [1,2], **Lili Zhang** [1], **Yuchao Song** [1], **Yonghua Li** [3], **Ye Zheng** [4], **Jinming Zhang** [4] and **Yongkun Cui** [4]

1   Institute of Applied Ecology, Chinese Academy of Sciences, Shenyang 110016, China;
    cuilei0121@163.com (L.C.); wuzj@iae.ac.cn (Z.W.); xueyanchina@163.com (Y.X.);
    Xiaofurong19@mails.ucas.ac.cn (F.X.); llzhang@iae.ac.cn (L.Z.); songyuchao@iae.ac.cn (Y.S.)
2   University of Chinese Academy of Sciences, Beijing 100049, China
3   North Huajin Chemical Industries Group Corporation, Panjin 124021, China; 13898988805@163.com
4   Jinxi Natural Gas Chemical Co., Ltd., Huludao 125001, China; hldjthgsb@126.com (Y.Z.);
    18642969003@163.com (J.Z.); cuiyongkun0123@163.com (Y.C.)
*   Correspondence: lidp@iae.ac.cn

**Abstract:** The application of nitrification inhibitors (NIs) is considered to be an efficient way to delay nitrification, but the effect of NIs combinations on soil nitrification and ammonia (NH$_3$) volatilization are not clear in soils with different pH values. In this study, we explored the effect of nitrapyrin (CP) and its combinations with 3, 4-dimethylepyrazole phosphate (DMPP), dicyandiamide (DCD) on the transformation of nitrogen, potential nitrification rate (PNR), and ammonia (NH$_3$) volatilization in a 120-day incubation experiment with three different pH values of black soil. Treatments included no fertilizer (Control), ammonium sulfate (AS), AS+CP (CP), AS+CP+DMPP (CP+DMPP), and AS+CP+DCD (CP+DCD). The application of NIs significantly decreased NO$_3^-$-N contents and potential nitrification rate ($p < 0.05$), while significantly increased NH$_4^+$-N contents ($p < 0.05$), especially CP+DCD and CP+DMPP were the most effective in the neutral and alkaline soils, respectively. In the acid soil, CP significantly increased total NH$_3$ volatilization by 31%, while CP+DCD significantly reduced by 28% compared with AS. However, no significant difference was found in NH$_3$ volatilization with and without NIs treatments ($p > 0.05$) in the neutral and alkaline soils. In conclusion, the combined nitrification inhibitors had the better efficiency in all three tested soils. CP+DCD and CP+DMPP are the most effective in inhibiting soil nitrification in the clay soils with higher pH value and lower organic matter, while CP+DCD had the potential in mitigating environment pollution by reducing N loss of NH$_3$ volatilization in the loam soil with lower pH value and higher organic matter. It provided a theoretical basis for the application of high efficiency fertilizer in different soils. Further studies under field conditions are required to assess the effects of these nitrification inhibitors.

**Keywords:** ammonium nitrogen; nitrate nitrogen; nitrification inhibitor; nitrification; ammonia volatilization

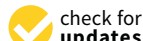

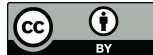

## 1. Introduction

Nitrate nitrogen (NO$_3^-$-N) is a negative ion and weakly adsorbed to the soil, and thus it tends to leaching [1]. NO$_3^-$-N leaching from farmlands is a major mechanism that causes the loss of N [2]. The high NO$_3^-$-N contents in drinking water are considered harmful to humans and animals and in surface water may cause eutrophication [3].

The application of nitrification inhibitors (NIs) with ammonium-based fertilizer is considered to be the most efficient way to reduce NO$_3^-$-N contents and mitigate environmental pollution through delaying nitrification. Nitrification inhibitors (NIs) delay the transformation of ammonium N (NH$_4^+$-N) to nitrate N (NO$_3^-$-N) and decrease N loss through inactivating the ammonia monooxygenase (AMO) enzyme, which is the key enzyme that results in the first, rate-limiting, step of nitrification [4]. NIs such as

nitrapyrin (CP), 3,4-dimethylepyrazole phosphate (DMPP), and dicyandiamide (DCD) are the most widely used in agricultural soils [5]. CP significantly increased retention of $NH_4^+$ and a lower accumulation of $NO_3^-$ [6], but CP is easily photolytic and volatile [7]. DMPP is effective at low application rates, has a low solubility in water (low leaching of DMPP), reduces the risk of nitrate leaching, and is not photolytic [8], but its price is too expensive. DCD has been proven to be effective in reducing nitrification rates and nitrate leaching, is non-volatile and low cost, but one of the major limitations of DCD is that it easily leaches out of the rooting zone, lowering its effectiveness, and the application rate is high [8,9]. Additionally, the mechanisms of NIs are various: CP is thought to exert an inhibitory effect by chelating the copper components of enzymes involved in ammonia oxidation [10], while two other NIs are directly binding and interact with ammonium monooxygenase (i.e., indiscriminate binding in the case of DMPP and blocking the electron transport in the cytochromes in the case of DCD) [11]. Moreover, previous studies have focused on the effects of the application of NIs alone on inhibiting nitrification [2,6,12]. There is little research about the effect of CP and its combinations with two other NIs in black soils with different soil properties. Therefore, it is essential to study the effect of CP and its combinations on decreasing $NO_3^-$-N contents for the efficient use of NIs in specific soils. The application of combined NIs will be cost-effective, which is more efficient in inhibiting nitrification.

NIs significantly reduced $NO_3^-$-N contents by delaying nitrification, and significantly increasing $NH_4^+$-N contents at the same time [13]. The higher $NH_4^+$-N contents in soils may increase the risk of ammonia volatilization [14], which is another main pathway of N loss in agricultural systems [15]. $NH_3$ volatilization plays an important role in N deposition, which is harmful to acidification in terrestrial systems, eutrophication of aquatic systems, and decline in biodiversity [1]. Sun et al. reported that CP increased $NH_3$ volatilization by 7.6–13% in an acid soil when the application rate of N is 180 kg N ha$^{-1}$ [16]. Additionally, other studies showed that DCD reduced $NH_3$ volatilization and DMPP had no impact on $NH_3$ volatilization [12,17]. Another study showed that soils with pH > 7.5 promoted the $NH_3$ volatilization [18]. Thus, it is clear that $NH_3$ volatilization is varied with different NIs and soil pH [19]. Hence, it is important to study the effect of different NIs treatments on $NH_3$ volatilization in soils with different pH values.

Black soil, namely Mollisol, is the main agricultural soil in northeast China [20]. In this study, an incubation experiment was carried out in three black soils with different pH values. The purpose of this study was to explore the effect of CP and its combinations on soil nitrification and $NH_3$ volatilization in black soils with different pH values, as well as to provide a theoretical basis for the application of NIs in the black soils of northeast China in the future.

## 2. Material and Methods

### 2.1. Soil Samples

Surface soil samples (0–20 cm depth) for incubation were collected from three sites: An acid soil at 853 farm (46°32′ N, 132°15′ E), Heilongjiang Province of China, a neutral soil at Nong'an (44°43′ N, 125°18′ E), Jilin Province of China, an alkaline soil at Da'an (45°31′ N, 123°56′ E), Jilin Province of China. The first two sampling sites were planted with maize and the last with rice (5 years ago, nearly 5 years in the form of abandoned dry land, no farming, no flooding), both of which were regularly fertilized.

At each site, surface soil was thoroughly mixed and immediately transported to the laboratory. The soils were air-dried, then passed through a 2 mm sieve to remove coarse plant debris and stones, finally they were stored at room temperature before use. Detailed physicochemical properties of the three soils are shown in Table 1.

**Table 1.** Physicochemical properties of three different soils.

| Soil Property | Acid Soil | Neutral Soil | Alkaline Soil |
|---|---|---|---|
| pH | $5.44 \pm 0.13$ | $7.66 \pm 0.07$ | $9.94 \pm 0.17$ |
| SOM (g/kg) | $52.25 \pm 1.91$ | $32.65 \pm 1.57$ | $30.12 \pm 0.54$ |
| Total C (g/kg) | $30.31 \pm 1.11$ | $18.94 \pm 0.91$ | $17.47 \pm 0.32$ |
| Total N (g/kg) | $2.63 \pm 0.03$ | $1.66 \pm 0.07$ | $0.92 \pm 0.15$ |
| $NH_4^+$-N (mg/kg) | $18.69 \pm 1.05$ | $27.83 \pm 3.46$ | $44.44 \pm 3.48$ |
| $NO_3^-$-N (mg/kg) | $80.68 \pm 1.46$ | $132.73 \pm 2.19$ | $24.33 \pm 2.16$ |
| Available P (mg/kg) | $48.40 \pm 2.13$ | $18.42 \pm 0.56$ | $15.43 \pm 0.32$ |
| Available K (mg/kg) | $401.45 \pm 34.27$ | $344.04 \pm 19.23$ | $375.28 \pm 24.33$ |
| Clay % | 12.3 | 37.3 | 60.6 |
| Silt % | 44.3 | 52.2 | 37.3 |
| Sand % | 43.4 | 10.4 | 2.1 |
| Texture class | loam | silt clay | clay |

SOM: Soil organic matter; Total C: Total carbon; Total N: Total nitrogen; $NH_4^+$-N: Ammonium nitrogen; $NO_3^-$-N: Nitrate nitrogen; P: Phosphorus; K: Potassium.

*2.2. Soil Incubation Experiment*

Soils were pre-incubated at 20% of their water holding capacity (WHC) for 1 week at $25 \pm 1$ °C. Additionally, 1 kg of dry soil was placed into a column (17 cm in diameter and 11.5 cm in height). Five treatments were applied with six replicates each during incubation experiment: Control: No fertilizer and NIs; AS: Ammonium sulfate; CP: Ammonium sulfate + nitrapyrin; CP+DMPP: Ammonium sulfate + nitrapyrin + 3,4-dimethylepyrazole phosphate; CP+DCD: Ammonium sulfate + nitrapyrin + dicyandiamide. N fertilizer was applied at a rate of $0.5$ g N $kg^{-1}$ dry soil. The application rates of CP, DMPP, and DCD were 0.5%, 1%, and 4% [21–23], respectively, on the w/w basis of N, and the application rate of each NI was reduced by 50% in the combination treatments. Soil samples were thoroughly mixed with corresponding fertilizers for each treatment and all columns were incubated in the dark at $25 \pm 1$ °C for 120 days. Deionized water was regularly added to maintain the maximum water-holding capacity (WHC) of the soil at 60%.

*2.3. Soil Sampling and Analysis*

During the incubation period, soil samples were taken from each treatment of three replicates at specific intervals (1, 3, 5, 7, 14, 21, 28, 35, 45, 70, 85, 100, 110, 120 days). All of the samples were stored at 4 °C for no more than 24 h for the determination of soil inorganic nitrogen and potential nitrification rate (PNR) (1, 7, 28, 70, 100 days).

Soil pH was determined using a ratio of 1:2.5 (*w/v*, soil/water) with a pH meter. Total C and total N of soil were determined using an Elemental Analyzer (Vario EL III, Hanau, Germany). Soil available phosphorus (AP) was determined by the molybdenum blue method on sodium bicarbonate extracts, soil available potassium (AK) was determined by extraction with ammonium acetate. Soil moisture content was determined by oven-drying at 105 °C for 8 h. Soil inorganic nitrogen ($NH_4^+$-N and $NO_3^-$-N) was extracted with 2 mol $L^{-1}$ KCl [24] and determined on a continuous flow analyzer (AA III, Norderstedt, Germany). The soil PNR was measured using the chlorate inhibition method [25].

$NH_3$ volatilization was measured from three other replicates by the venting method [26]. The chambers, made of polyvinyl chloride rigid plastic column (15 cm internal diameter, 10 cm high), were pushed approximately 5 cm into the soil. Two circular pieces of sponge with a thickness of 2 cm and a diameter of 15 cm were uniformly impregnated with glycerophosphate (15 mL of 5% phosphoric acid in 4% of glycerol solution) and placed in a chamber. The lower sponge is 5 cm from the bottom of the column pot and the upper sponge is flat with the top of the column. The upper sponge was to absorb ammonia from the air and prevent contamination, the lower sponge was to absorb ammonia from within the device and the ammonia volatilized from the soil. The experimental device was shown in Figure 1. When sampling, the lower sponge was taken out, quickly put it into a plastic bag, and sealed. Meanwhile, a freshly soaked glycerophosphate sponge was put in place.

The upper sponge, depending on its wet and dry situation, was replaced every 3–7 days. Finally, the lower sponges were put into 500 mL plastic bottles respectively, and 300 mL 1 mol·L$^{-1}$ KCl solution was added to make the sponges completely immersed. After 1 h of oscillation, the ammonium nitrogen in the leaching solution was analyzed by a continuous flow analyzer. Gas samples were collected every 2 days in the first week, and then every 7 days in the first month, finally every 10 days in the other months. NH$_3$ was collected a total of nine times during 45 days.

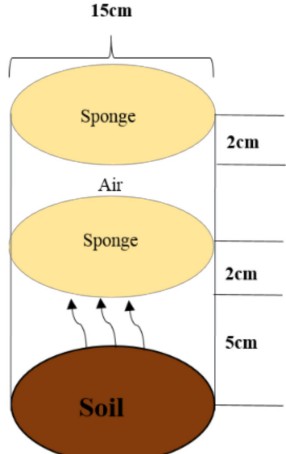

**Figure 1.** Venting method for determination of ammonia volatilization.

*2.4. Statistical Analysis*

Data were presented as the means of the three replicates. Analysis of variance (ANOVA) was conducted to test the effects of all treatments at each incubation time. Significant differences between the means were analyzed using Tukey's multiple comparisons at the 0.05 probability level. All statistical analyses were conducted using the statistical software SPSS 22.0. Graphs were prepared using Origin 9.0.

## 3. Results

*3.1. Soil Inorganic Nitrogen Content*

The contents of NH$_4^+$-N differed over the incubation period due to the differences in soil pH and the use of NIs (Figure 2). Within the Control in the acid and alkaline soils, the contents of NH$_4^+$-N remained almost unchanged at <31 mg kg$^{-1}$ soil, while the Control in the neutral soil had NH$_4^+$-N > 31 mg kg$^{-1}$ soil during the first 21 days of incubation (Figure 2a–c). In the AS, the NH$_4^+$-N contents initially increased rapidly, and then declined with the first 14 days in all tested soils. Additionally, the NH$_4^+$-N contents kept higher in the acid soil than in the two other tested soils until the end of the incubation time. In the NIs, a sharp increase in NH$_4^+$-N contents was observed during 7 days, but after that, NH$_4^+$-N contents were gradually declined but significantly higher than AS (Figure 2a–c). Moreover, NIs were more effective in inhibiting nitrification in both neutral and alkaline soils (Figure 2, $p < 0.05$). After 120 days of incubation, all NIs still had higher NH$_4^+$-N contents in both the acid and alkaline soils, but had no significant difference with AS in the neutral soil.

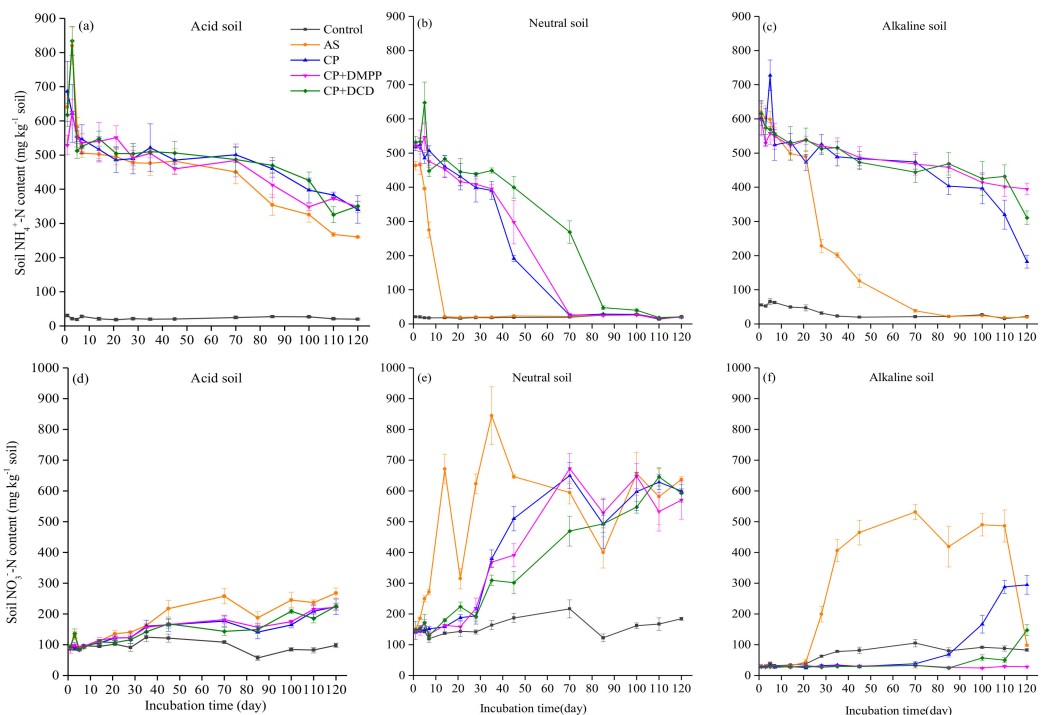

**Figure 2.** Changes in ammonium nitrogen (NH$_4^+$-N) and nitrate nitrogen (NO$_3^-$-N) in the acid (**a**,**d**), neutral (**b**,**e**), and alkaline soil (**c**,**f**) among different treatments during 120 days of incubation. Treatments: Control: No fertilizer and NIs; AS: Ammonium sulfate; CP: Ammonium sulfate + nitrapyrin; CP+DMPP: Ammonium sulfate + nitrapyrin + 3,4-dimethylepyrazole phosphate; CP+DCD: Ammonium sulfate + nitrapyrin + dicyandiamide. Error bars represent standard deviations (n = 3). The same as below.

The inhibition of nitrate production by inhibitor treatments varied with soils with different pH values. NO$_3^-$-N contents gradually increased in Control and AS soil samples (Figure 2d–f, $p < 0.05$), with less NO$_3^-$ being produced in the acid soil than the two other soils (Figure 2d–f). The production of NO$_3^-$ was significantly reduced when NIs were added with N fertilizer, particularly lasting over 110 days in the acid and alkaline soils, while only 70 days in the neutral soil (Figure 2d–f, $p < 0.05$). The results of two-way ANOVA also indicated that different treatments, soils, and their interaction had a significant influence on NH$_4^+$-N contents and NO$_3^-$-N contents (Table 2, $p < 0.05$).

**Table 2.** Two-way ANOVA ($p < 0.05$) examining the effect of different treatments (T), soils with different pH vales (S), and their interaction (T * S) on NH$_4^+$-N, NO$_3^-$-N, potential nitrification rate (PNR), and ammonia volatilization (NH$_3$) during the incubation.

| Factors | DF | NH$_4^+$-N | | NO$_3^-$-N | | PNR | | NH$_3$ | |
|---|---|---|---|---|---|---|---|---|---|
| | | F | $p$ | F | $p$ | F | $p$ | F | $p$ |
| T | 4 | 172.5 | *** | 10.1 | *** | 18.0 | *** | 2.3 | n.s. |
| S | 2 | 21.8 | *** | 66.3 | *** | 37.9 | *** | 39.8 | *** |
| T * S | 8 | 3.9 | *** | 4.4 | *** | 12.6 | *** | 1.7 | n.s. |

n.s.: Not significant; *** significant at $p < 0.001$.

### 3.2. Soil Potential Nitrification Rate (PNR)

Different treatment combinations and soils with different pH values significantly affected soil PNR, and the impact of their interaction was also significant from a statistical point of view (Table 2, $p < 0.05$). The results of soil PNR in each treatment of three soils during the incubation were shown in Figure 3. PNR was the highest in the neutral soil than in the acid and alkaline soils. In the acid soil, the highest value of PNR was found in the Control during the entire period of incubation (Figure 3). No significant difference

was found among all of the treatments on day 1 (Figure 3a, $p < 0.05$). CP and CP+DCD had significantly lower PNR than AS on days 7 and 28 ($p < 0.05$), while no significant difference was found between CP+DMPP and AS. However, the PNR value in AS was obviously lower than in CP+DMPP and CP+DCD on days 70 and 100 ($p < 0.05$), while no significant difference was found between CP and AS ($p < 0.05$). In the neutral soil, the PNR in Control was higher than in AS on days 1 and 7 (Figure 3b, $p < 0.05$), while on days 28 and 70, AS had significantly higher than the Control, but no significant difference was found between the Control and AS on day 100. All of the treatments with NIs significantly reduced PNR on each sampling day during the entire period of incubation (Figure 3b, $p < 0.05$). No significant difference was found among all treatments amended with NIs on days 7, 28, and 100 ($p < 0.05$), although CP+DCD had significantly lower than CP+DMPP on days 70 ($p < 0.05$). In the alkaline soil, AS had the highest PNR value on days 28, 70, and 100 (Figure 3c, $p < 0.05$), while no significant difference was found between the Control and AS on days 7 and 28. CP and CP+DMPP significantly reduced PNR during the entire period of incubation ($p < 0.05$). However, CP+DCD only reduced PNR on days 1, 28, and 70 ($p < 0.05$).

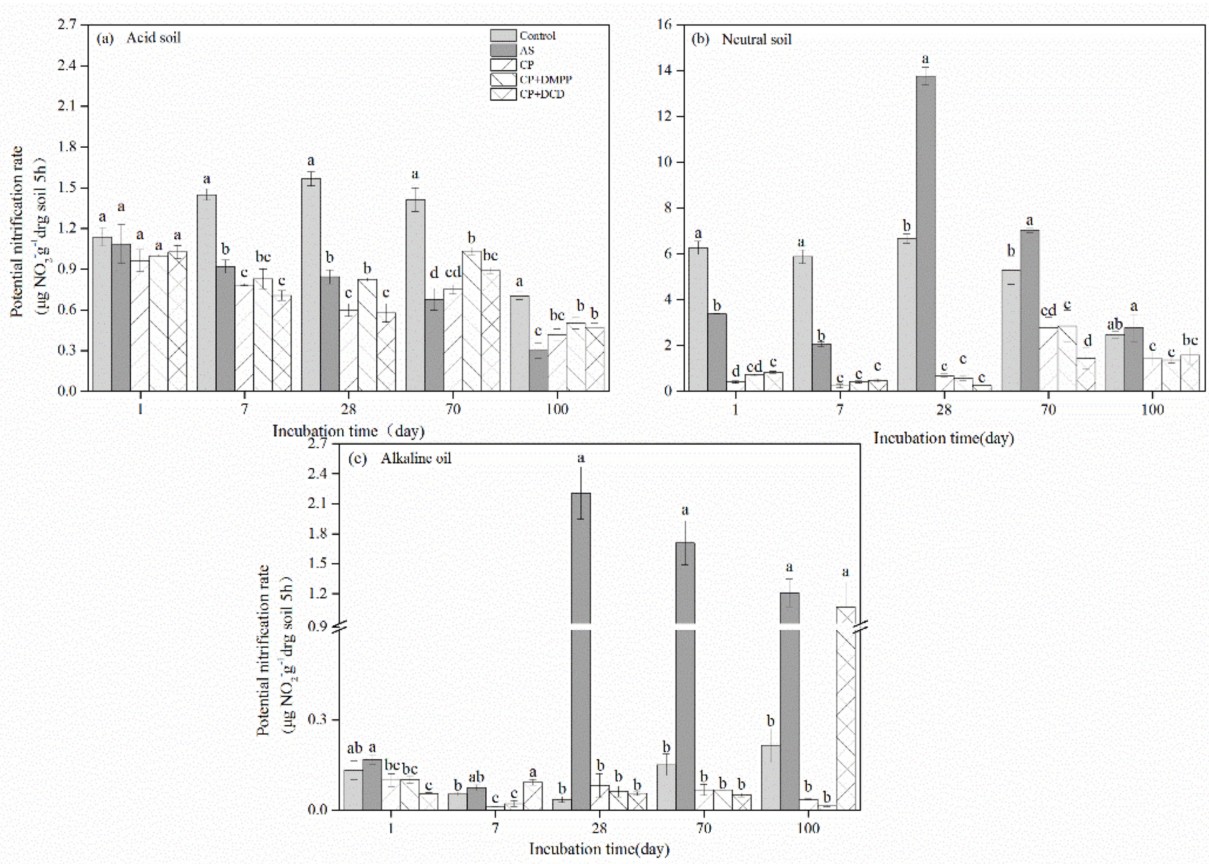

**Figure 3.** Potential nitrification rate (PNR) in the acid (**a**), neutral (**b**) and alkaline soils (**c**) under different treatments on each sampling day during the incubation. Error bars represent standard deviations (n = 3). The different letters above the figures of the same sampling day indicate significant differences between treatments at $p < 0.05$ by Tukey's test.

### 3.3. Soil $NH_3$ Volatilization

The dynamics of the rate of ammonia volatilization of three soils were shown in Figure 4. In all of the three soils, the lowest value of $NH_3$ volatilization was observed in the acid soil, while the highest value in the alkaline soil, which reached 10.06 mg m$^{-2}$ h$^{-1}$ (Figure 4A,C, $p < 0.05$). $NH_3$ volatilization of Control remained almost unchanged at less than 0.5 and 1 mg m$^{-2}$ h$^{-1}$ on each sampling time during the incubation experiment in both the neutral and alkaline soils, respectively, while the Control had a similar pattern

with the treatments with N fertilizer in the acid soil (Figure 4). No significant difference was found among all of the treatments with N fertilizer in all three soils after 45 days of incubation. In the acid soil, no significant difference was found among all of the treatments on days 5, 14, 28, and 45, and the NIs treatments also had no significant difference on days 7 and 21 (Figure 4A, $p < 0.05$). However, CP+DMPP had significantly higher ammonia volatilization value on days 1 and 3, while no significant difference was found between CP and CP+DCD. For 35 days, the highest value in ammonia volatilization was detected in CP, but no significant difference was found between CP+DMPP and CP+DCD (Figure 4A, $p < 0.05$). In both the neutral and alkaline soils, no significant difference was found between all of the treatments with the N fertilizer (Figure 4B,C, $p < 0.05$), and the rate of ammonia volatilization declined with the incubation time.

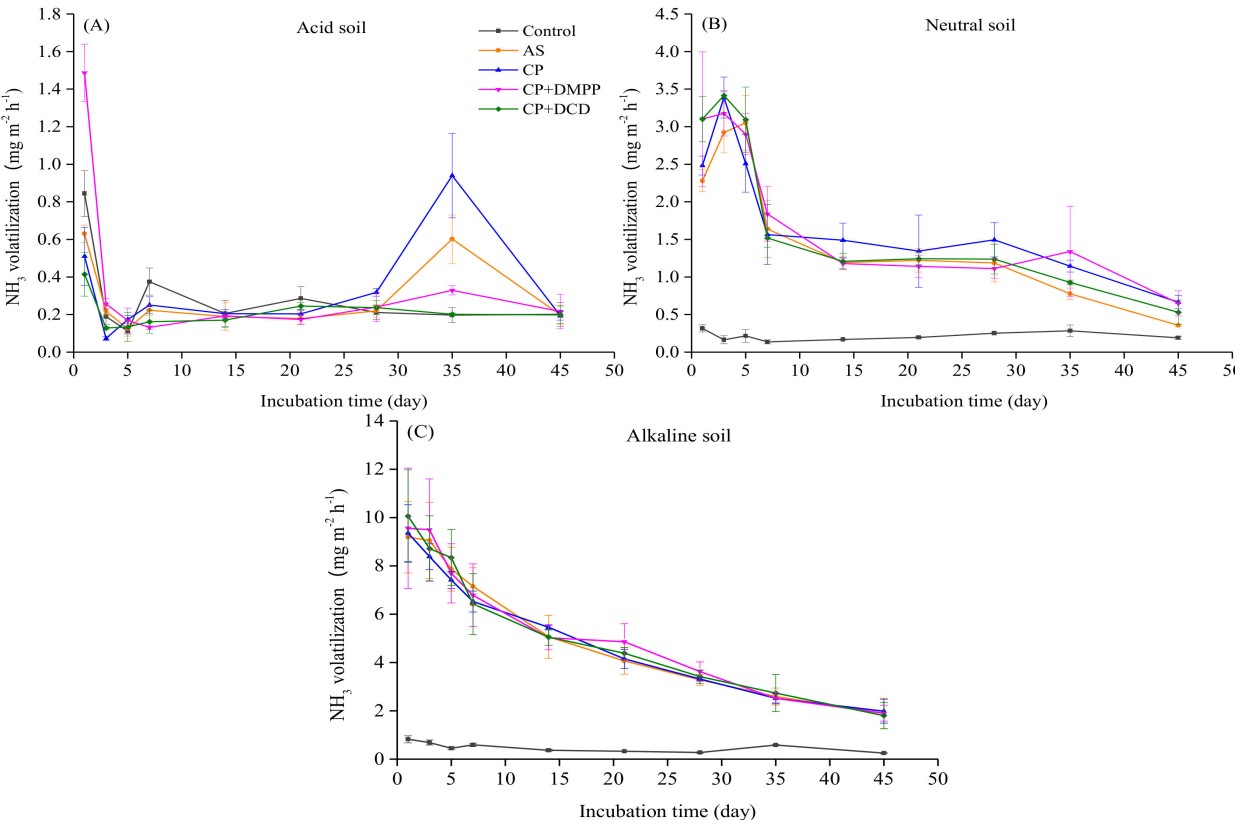

**Figure 4.** Dynamics of the rate of ammonia volatilization of different treatments in the acid (**A**), neutral (**B**), and alkaline soils (**C**) under each sampling day during the incubation. Error bars represent standard deviations (n = 3).

Figure 5 showed the accumulative $NH_3$ volatilization of different treatments in three soils. In the acid soil, the accumulative $NH_3$ volatilization had no significant difference between the Control and AS. CP was significantly higher, while CP+DCD was significantly lower in $NH_3$ volatilization compared with AS (Figure 5, $p < 0.05$). In the both neutral and alkaline soils, the accumulative $NH_3$ volatilization considerably increased in AS and NIs compared with the Control, but no significant difference was found between AS and NIs (Figure 5, $p < 0.05$). All of the treatments were the highest in the alkaline soil, followed by the neutral soil, and the lowest value of $NH_3$ volatilization was found in the acid soil (Figure 5, $p < 0.05$).

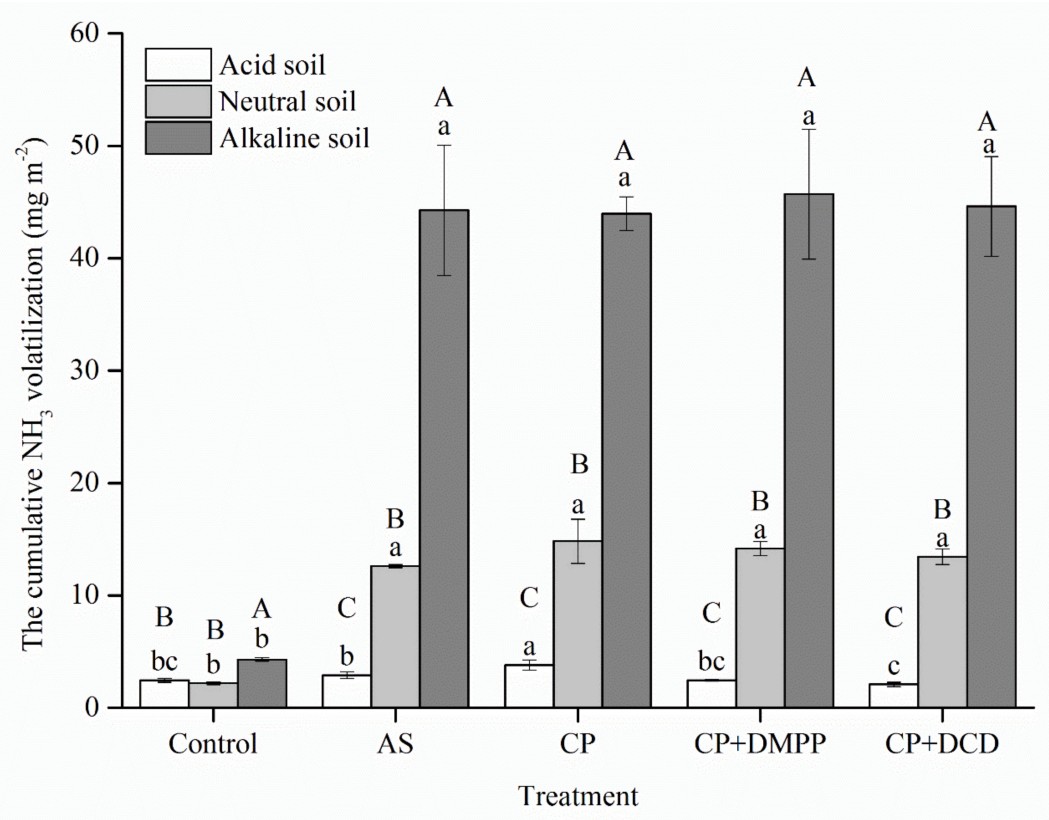

**Figure 5.** The cumulative ammonia volatilization of different treatments in the acid, neutral, and alkaline soils. Error bars represent standard deviations (n = 3). The different lowercase letters above the figures indicate significant differences between different treatments within the same soil at $p < 0.05$ by Tukey's test; the different capital letters above the figures indicate significant differences between different soils in the same treatment at $p < 0.05$ by Tukey's test.

## 4. Discussion

In all of the three tested soils, the decrease in $NH_4^+$-N contents was accompanied by the increase in $NO_3^-$-N contents in AS, indicating the occurrence of nitrification during the incubation period. Moreover, the results of the Pearson correlation analysis revealed that soil $NH_4^+$-N contents had a significantly negative correlation with soil $NO_3^-$-N contents (Table 3, $p < 0.01$). The result is in line with a previous study, which indicated that in a 28-day microcosm incubation, the addition of $NH_4NO_3$ significantly increased the $NO_3^-$-N concentrations over time, which coincided with the dramatic declines in exchangeable $NH_4^+$-N [27]. Additionally, our study showed that the use of all NIs (CP, CP+DMPP, CP+DCD) with ammonium sulfate significantly improved $NH_4^+$-N contents and reduced $NO_3^-$-N contents by reducing soil nitrification for all of the three soils. The phenomenon is consistent with a previous study, which showed that DMPP significantly inhibited nitrification in all of the three contrasting soils in an incubation experiment [28]. Additionally, CP inhibited nitrification in four agricultural soils in an incubation experiment [6], and DCD delayed nitrification in two soils with different pH values in a pot experiment [29]. However, no significant difference was found in inorganic nitrogen among NIs in the acid soil, while CP+DCD kept higher $NH_4^+$-N contents for a longer period in the neutral soil and CP+DMPP in the alkaline soil (Figure 2). The reason for this result may be the weak soil nitrification in the acid soil and the low potential nitrification rate (PNR) in both the neutral and alkaline soils after adding NIs into the N fertilizer [30,31]. The PNR is an index which aims to determine the maximum capacity of nitrifiers to transform ammonium into nitrate [32]. The results of Pearson correlation suggested that PNR showed significant and negative correlations with $NH_4^+$-N, but significantly positive correlations with $NO_3^-$-N (Table 3, $p < 0.01$). All of the treatments with NIs significantly decreased PNR in the three soils, especially in both neutral and alkaline soils (Figure 3), suggesting that NIs were more

effective in inhibiting nitrification in these two soils [33]. In addition, the highest PNR was found in the neutral soil. Gong et al. also reported that PNR was higher in the soils with higher pH [29].

**Table 3.** Pearson correlation analysis between $NH_4^+$-N, $NO_3^-$-N, potential nitrification rate (PNR), and ammonia volatilization ($NH_3$).

| Item | $NH_4^+$-N | $NO_3^-$-N | PNR | $NH_3$ |
|---|---|---|---|---|
| $NH_4^+$-N | 1 | −0.321 ** | −0.538 ** | 0.123 |
| $NO_3^-$-N | | 1 | 0.757 ** | −0.214 * |
| PNR | | | 1 | −0.159 |
| $NH_3$ | | | | 1 |

** Correlation is significant at the 0.01 level (two-tailed); * correlation is significant at the 0.05 level (two-tailed).

The differences in the efficiency of NIs among the three soils may be attributed to the effect of the contrasting physicochemical in three soils. One of these properties was soil pH, which was considered as one of the most important factors affecting NI efficiency, since the pH value could impact the mobility and degradation of NIs in soils [30]. The results of Pearson correlation analysis indicated that soil pH had a negative correlation with soil $NO_3^-$-N (Table 4, $p < 0.05$). Yang et al. also showed that DCD was more efficient in the neutral soil, while DMPP was efficient in the alkaline soil [34]. A meta-analysis also showed that NIs were more effective in the neutral soil (pH 6.0–8.0) and in alkaline soil (pH ≥ 8.0), but not in acid soil (pH ≤ 6.0) [35]. It seems that the nitrification activity in soils with higher pH is generally higher, which should be conducive to the inhibition effect of nitrification inhibitors [36]. Aside from soil pH, soil organic matter (SOM) and soil texture have also been recognized as the main factors influencing the effectiveness and the persistence of NIs [29,37]. The adsorption of SOM on NIs reduces its mobility, volatility, bioactivity, and thus their effectiveness, but increases the persistence of NIs due to the sorption of NIs by the SOM [38,39]. $NH_4^+$-N contents were kept higher in the acid soil during the entire period of incubation than in the two other soils, since the content of SOM in the acid soil (52.25 g $kg^{-1}$) was higher than those in the two other tested soils (32.65 g $kg^{-1}$; 30.12 g $kg^{-1}$), which increased the persistence of NIs. DCD also had higher $NH_4^+$-N contents in a red soil with higher SOM [30]. Additionally, our study indicated that CP+DMPP was more effective in inhibiting nitrification in the alkaline soils with lower SOM, but not in the two other tested soils. It could be inferred that the adsorption of DMPP by high SOM [9], resulted in low availability in the two other tested soils with higher SOM [40]. Moreover, the higher $NH_4^+$-N contents in soils may be due to the retention of ammonium ions, which usually adhere to soil organic matter (SOM), thus reducing nitrate leaching [1,41]. The contents of $NH_4^+$-N decreased gradually with the incubation time, which might be contributed to nitrification (caused by decomposition of NIs), $NH_3$ volatilization [1], and microbial immobilization [42]. In this study, NIs markedly retarded the soil nitrification in the three tested soils with different soil textures, especially in the latter two tested soils (Figure 1). This is in line with the efficacy of DMPP reported in two soils with different textures in an incubation experiment [40]. However, distinct differences in the extent and duration of NIs effect among the soils were observed. The results of Pearson correlation analysis also indicated that soil texture had a significantly negative correlation with soil $NO_3^-$-N (Table 4, $p < 0.05$). In the silt clay and clay soils, NIs delayed $NH_4^+$ oxidation more efficiently than in the loam soils, which is not in line with previous studies. Previous studies demonstrated that nitrification was less inhibited in the soils with higher clay and silt content, where DMPP may have been absorbed [40]. The main reason for this discrepancy is the higher pH and lower SOM in the tested soil [38,43].

**Table 4.** Pearson correlation test between soil properties (pH, soil organic matter (SOM), and texture), $NH_4^+$-N, $NO_3^-$-N, and ammonia volatilization ($NH_3$).

| Item | $NH_4^+$-N | $NO_3^-$-N | $NH_3$ |
|---|---|---|---|
| Soil pH | −0.018 | −0.217 * | 0.523 ** |
| SOM | 0.064 | 0.089 | −0.452 ** |
| Soil texture | −0.013 | −0.231 ** | 0.530 ** |

** Correlation is significant at the 0.01 level (two-tailed); * correlation is significant at the 0.05 level (two-tailed).

The sharp increase in $NH_3$ volatilization was clearly observed in the early days after the application of N fertilizer and NIs during the incubation period in three soils, which could be due to higher $NH_4^+$ [44]. $NH_3$ volatilization was higher in the neutral soil (pH: 7.66) and alkaline soil (pH: 9.94) than in the acid soil (pH: 5.44). The reason for the result was that ammonia emission was a physico-chemical process that occurred under alkaline soil when the soil pH is high ($\geq$7.6) [45]. Additionally, the results of Pearson correlation analysis demonstrated that soil pH had a significantly positive correlation with ammonia volatilization (Table 4). The gradual decrease in $NH_3$ volatilization with the incubation time might be due to the fixation of ammonium [46]. The highest $NH_3$ volatilization was found in the alkaline soil, which indicated that $NH_3$ volatilization is a major N loss pathway in the alkaline soil [47]. The magnitude of $NH_3$ volatilization is affected by many soil and environment factors, especially NIs and soil pH [48]. Many studies have demonstrated that NIs increased $NH_3$ volatilization [19,49], while others reported that no significant differences were detected, as well [46]. For instance, DCD increased $NH_3$ volatilization by 7% [50] and DMPP had no influence [14]. Additionally, DCD increased $NH_3$ volatilization in an acid soil by 4–16% (pH: 5.9) [51], but reduced $NH_3$ volatilization by 20.61–41.51% in an alkaline soil (pH: 7.84) [52]. In our study, the application of CP increased total $NH_3$ volatilization by 31%, while the combined NIs CP+DCD reduced by 28%, CP+DMPP had no impact on total $NH_3$ volatilization in the acid soil (Figure 5). The increase in $NH_3$ emission in NI treatments with low pH soil (pH: 5.4) was also observed [53], which may be explained by the fact that $NH_3$ emission can occur at soil pH as low as 5.5 when a large amount of $NH_4^+$ is applied [54]. A previous study has also shown that CP increased soil pH [55,56], which could lead to higher $NH_3$ volatilization [57]. However, all of the NIs treatments had no impact on total $NH_3$ volatilization in the two other tested soils (Figure 5). The result is consistent with a field study which also found no effect of NIs addition on cumulative $NH_3$ emission [58]. Some studies indicated that $NH_3$ volatilization increased, as a result of the rise of pH [59,60], which was similar with our result. The order of the total $NH_3$ volatilization in each treatment was alkaline soil (pH: 9.94) > neutral soil (pH: 7.66) > acid soil (pH: 5.44) (Figure 5). These findings are in accordance with the results of Pearson correlation analysis, which showed a soil pH significant positive correlation between the soil pH and ammonia volatilization (Table 4, $p < 0.01$). This phenomenon suggested that the lower nitrification rates (caused by NIs) may reduce soil acidification, leading to a prolongation of pH rise in alkaline soil, and resulting in increased $NH_3$ volatilization [45].

## 5. Conclusions

All of the NIs treatments effectively inhibited nitrification in the three soils with different properties. No significant difference in inhibiting nitrification was detected in the acid soil among all of the NIs treatments. While combined NIs were more efficient in inhibiting nitrification in the clay soils with higher pH and lower organic matter, CP+DCD was more efficient in the neutral soil and CP+DMPP was the most effective in the alkaline soil. All of the NIs treatments significantly decreased PNR in the clay soils with higher pH and lower organic matter during the incubation time. The highest PNR was in the neutral soil, followed by alkaline soil and the lowest was in the acid soil. CP increased $NH_3$ volatilization, but CP+DCD reduced $NH_3$ volatilization in the loam soils with lower pH and higher organic matter. No significant difference was found in $NH_3$ volatilization with NIs or without NIs in the clay soils with higher pH and lower organic matter. Therefore, we

proposed that CP+DCD and CP+DMPP were more efficient in inhibiting nitrification in the neutral and alkaline soils, respectively. In addition, CP+DCD had the potential to mitigate environmental pollution by reducing $NH_3$ volatilization in the acid soil. In conclusion, the application of combined nitrification inhibitors with nitrogen fertilizer is a cost-effective way, which can reduce the frequency and amount of nitrogen fertilizer applied, improve the availability of nitrogen, and thus improve the nitrogen use efficiency (NUE). Further field studies to verify the effect are needed.

**Author Contributions:** Conceptualization, L.C. and D.L.; formal analysis, L.C.; funding acquisition, D.L. and Z.W.; investigation, D.L.; methodology, L.C.; resources, Y.X., F.X., L.Z., Y.S., Y.L., Y.Z., J.Z. and Y.C.; supervision, L.C., D.L. and Z.W.; writing—original draft, L.C.; writing—review and editing, L.C. and D.L. All authors have read and agreed to the published version of the manuscript.

**Funding:** This research was funded by the National Key Research and Development Program of China, grant number 2017YFD0200707.

**Institutional Review Board Statement:** Not applicable.

**Informed Consent Statement:** Not applicable.

**Data Availability Statement:** All relevant data is contained within the article.

**Conflicts of Interest:** The authors declare no conflict of interest.

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
