# Peer review of "Effects of Nitrification Inhibitors on Soil Nitrification and Ammonia Volatilization in Three Soils with Different pH"

_agronomy, doi:10.3390/agronomy11081674_

Round 1
Reviewer 1 Report
The manuscript has been improved. Thanks!
Several grammatical errors in the new text. Please correct grammar!
Line 23-24 In conclusion, the combined nitrification inhibitors had the better
efficiency in all three tested soils.
What does it mean "the better"? Please be specific.
Line 326: This is in line with the efficacy of, which was be reported for two soils with different texture in an incubation. Please correct "which was be reported"
Line 200: Different treatment combinations and soils with different pH significantly affected soil PNR, and the impact of their interaction was also significant from statis point of view.
What is "statis"?
57 : Moreover, previous studies have focused the effects of the application of NIs alone on inhibiting nitrification [2, 6, 12].
Consider "focused on NI application effect on nitrification inhibition only".
Reviewer 2 Report
The authors already revised the paper properly. It can be accepted now.
Author Response
Please see the attachment.

This manuscript is a resubmission of an earlier submission. The following is a list of the peer review reports and author responses from that submission.
Round 1
Reviewer 1 Report
The manuscript "Effects of Nitrification Inhibitors on Soil Nitrification and Ammonia Volatilization in Three Soils with Different pH" concentrates to study effects of combination of three different nitrification inhibitors on nitrification and ammonia volatilization. Results of this kind of study may help to find the most effective ways to decrease nitrogen losses from different agricultural soils as dissolved and gaseous forms, which is very important when trying to mitigate e.g. surface water eutrophication. However, previous studies on the effects of different nitrification inhibitors are not scarce and therefore novelty value of this study is not very high. Nevertheless, the study could be a nice addition to the existing literature and interesting for the readers of the journal.
I can’t recommend publication of it in the form in which it is now.
- One of the biggest problems is that the soils used in the study differ not only by their pH but also many other properties. For example, soil textual classes vary a lot, the acid soil being clearly coarser than the other two soils. Despite this, the authors seem to conclude that soil pH is the only factor that affects differences between the soils. It may well be that pH is the most predominant factor, but the authors should also thoroughly consider the effects of the other soils properties on the results as well.
- In the Introduction, it should be more clearly justify why combination of different NIs was studied. I also suggest that the abundance of AOA and AOB mentioned in the end of Introduction should be explained and connected more clearly to the background and the aims of the study.
- Description of the experimental part is not exact enough. It is not explained how the amount of added N fertilization was decided (it seems that the added amount was very high). It’s also unclear what were the amounts of nitrification inhibitors used in the experiment and how the amounts were decided. Were the amounts of NIs different in different treatments? If so, please justify why (and in Discussion, consider how the different amounts may have affected the results). I also suggest the authors to consider how can they be sure that the subsamples taken for inorganic N and other analyses during the experiment were homogeneous and representative enough.
- Results chapter needs much clarification; text contains several inaccuracies and flaws when referring to figures or different soils (not only technical but also interpretation flaws or inaccuracies); this chapter should be rewrite particularly carefully. In addition, Figures 2 and 4 are hard to interpret because their size is too small, and results of statistical analyses are not presented in the figures.
- Discussion stays on too shallow level, is in places vague and do not offer much novel scientific information. I suggest the authors to consider not only pH but also other soil properties as explaining factors for the differences between the soils. Technical flaws in Discussion should be corrected as well (e.g. p-values mentioned in the text differ from those mentioned in the table).
- Overall, the manuscript is not carefully written and therefore the whole text and language needs throughout revision. For example, words seem to be missing from some sentences, here and there tenses should be corrected, some of the abbreviations (at least AOA, AOB, NUE) are not explained and so on.
Taking into account my comments above, my opinion is that the manuscript does not fill the criteria set for high quality scientific article in its current form. Rewriting the manuscript will require quite a lot of effort but if the authors are able to overcome the problems mentioned above and substantially improve the manuscript, I suggest resubmission of it.
Reviewer 2 Report
It is an interesting study, especially with the large number of treatments, replications, and long time of soil incubations. The manuscript is suitable for publication in the MDPI Agronomy Journal.
The main problem is difficulty separating nitrification rates of applied NH4 fertilizer and the ammonia that comes from mineralization of soil organic matter. The confounding effect of these two processes making difficult to conclude about the impact of the nitrification inhibitor and their combinations on nitrification rate of fertilizer nitrogen, soil nitrogen and ammonia volatilization. One solution would be to consider a new Figure similar, to Figure 4 or 5, where the nitrification rates for the soil organic matter are subtracted.
In addition, the alkaline soil in this study came from the rice paddy that likely has different soil microbial communities, thus, making comparison the alkaline soil to the acid and neutral challenging.
Round 2
Reviewer 1 Report
Thank you for the revised manuscript. Content of the manuscript has been somewhat clarified, but unfortunately, there are still some shortcomings and especially Discussion section is still partly vague, unclear and on pretty shallow level.
One significant shortage is that even if the authors seem to be aware that NIs may affect soil pH and they consider how pH changes during the incubation (e.g. due to lower nitrification rates which may reduce soil acidification as they mention) may have affected the results, pH was not measured during the incubation but only in the beginning of the study. There is also some overinterpretation of the results; it is said e.g. that ‘CP+DCD and CP+DMPP are the most effective in inhibiting nitrification in the neutral and alkaline soil’, but actually those treatments differed significantly from the other NI treatments only occasionally (especially CP+DMPP) and no clear trends were seen. In acid soil, it was found that some of the treatments decreased and some increased NH3 volatilization rates but taking into account the comparatively low NH3 volatilization rates in acid soil, practical importance of this finding may not be very high, especially considering that amount of added N-fertilization seemed to be extremely high; in field conditions with lower N-fertilization rates, NH3 emissions would probably be clearly lower in the acid soil than they were in the experimental conditions. Additionally, the finding that volatilization of NH3 was highest in the soil with highest pH and lowest in the acid soil, is not brand new scientific information. I’m also wondering why possible effects of N mineralization on the results were not considered at all.
I understand that the effect of soil pH was the main aim of the study, but since also other soil properties varied, it should have been taken into account in the discussion as I suggested in my previous comments. In addition to soil pH, the authors raise only SOM content of soil as another explaining factor for the observed differences between the soils, but it stays more or less unclear what the actual affecting mechanism of SOM is supposed to be. It’s mentioned that SOM may adsorb ammonium ions, but it’s confusing that the authors call this process fixation of ammonium which actually refers to adsorption of ammonium ions into inter-layer sites of clay minerals. Since clay content of soils seemed to vary a lot, it would have been beneficial to consider at least the effect of soil texture on the results because clay minerals may act as adsorbing surfaces for ammonium ions as well as SOM does.
I also find it problematic that the authors mention that their results are in line with previous studies, but they do not mention anything about the conditions in which previous studies have been conducted. Therefore, it’s impossible for the reader to know whether those studies and their results are comparable to the results of the present study (even if the reader does not read through all the references mentioned).
Finally, although the language of the manuscript has been somewhat improved, it would still have required further improvement in many places. In addition, there are still few flaws when referring in the text to soils or days which indicates that rewriting of the manuscript has not been done very carefully. If the authors are going to submit the manuscript to another journal, I strongly suggest them once again checking content of the text and to let a native English speaker to check the language before submission.
I understand that the authors will be disappointed because they have put much effort for conducting the study and writing and revising the manuscript, but unfortunately, taking into account the shortcomings of the manuscript, I cannot recommend publication of it.